# From 2-D to 0-D Boron Nitride Materials, The Next Challenge

**DOI:** 10.3390/ma12233905

**Published:** 2019-11-26

**Authors:** Luigi Stagi, Junkai Ren, Plinio Innocenzi

**Affiliations:** Laboratorio di Scienza dei Materiali e Nanotecnologie, CR-INSTM, Dipartimento di Chimica e Farmacia, Università di Sassari, Via Vienna 2, 07100 Sassari, Italy; lstagi@uniss.it (L.S.); j.ren@studenti.uniss.it (J.R.)

**Keywords:** boron nitride, 2D materials, quantum dots, fluorescence, nanocomposites

## Abstract

The discovery of graphene has paved the way for intense research into 2D materials which is expected to have a tremendous impact on our knowledge of material properties in small dimensions. Among other materials, boron nitride (BN) nanomaterials have shown remarkable features with the possibility of being used in a large variety of devices. Photonics, aerospace, and medicine are just some of the possible fields where BN has been successfully employed. Poor scalability represents, however, a primary limit of boron nitride. Techniques to limit the number of defects, obtaining large area sheets and the production of significant amounts of homogenous 2D materials are still at an early stage. In most cases, the synthesis process governs defect formation. It is of utmost importance, therefore, to achieve a deep understanding of the mechanism behind the creation of these defects. We reviewed some of the most recent studies on 2D and 0D boron nitride materials. Starting with the theoretical works which describe the correlations between structure and defects, we critically described the main BN synthesis routes and the properties of the final materials. The main results are summarized to present a general outlook on the current state of the art in this field.

## 1. Introduction

Boron nitride (BN) is a chemically stable material exhibiting four different polymorphs: hexagonal (h-BN), cubic (c-BN), rhombohedral (r-BN), and wurtzite (w-BN) [1,2,3,4,5]. The different BN allotropes have attracted considerable interest as a possible alternative to diamond for their great hardness and high thermal stability [3,4,5,6]. Despite a thermal stability higher than diamond (1473 K) and a low reactivity with steel, c-BN still presents a more moderate hardness in comparison with diamond (H_V_ = 40–60 GPa) [3]. Nevertheless, C_2_-BN nanocomposites (c-BN: diamond, 1:2) can reach a very high hardness, H_V_ = 85 GPa. More recently, w-BN displayed interesting hardness properties, comparable with the other allotropes [5]. Pure w-BN, treated at a pressure of 10–20 GPa and high temperature (400–1900 °C) reached a hardness as high as H_V_ = 46 GPa. A two-stage shear mechanism is responsible for the unusual hardness of w-BN [6].

h-BN presents an sp^2^ hybridization of B–N bonds, and it crystallizes in a layered structure like graphene (Figure 1). While B and N atoms are held together with strong covalent bonds densely covering the plane, the different BN layers interact via Van der Waals forces. Contrary to graphene, B–N is strongly polarized with electronegative N and almost vacant B. BN nanoplates are known to preserve their thermal stability up to 1000 °C, undergoing oxidation in the range between 1000–1200 °C, with the formation of B_2_O_3_. For thin BN nanosheets the oxidation process starts at 850 °C [7,8,9,10]. h-BN monolayers present a thermal conductivity of 751 W mK^−1^ at 25 °C conjugating high thermal conductivity with electrical insulation as a good candidate for heat dissipation in future electronic devices [11].

Optical spectroscopy measurements on h-BN crystals have demonstrated evidence for an indirect band gap of 5.955 eV [12]. More recently, first principle calculations described the evolution of band gap as a function of the numbers of layers, predicting the crossover to direct band gap at a limit of one layer [13], later verified experimentally by Elias et al. [14].

Boron and nitrogen show the ability to create ordered superstructures under ultra-high vacuum conditions. Borazine, widely used in film depositions, has the tendency to decompose on Ru and Rh substrates, forming a self-assembled regular mesh [15,16,17]. The derived structure presents hexagonal pores at distances of 3.2 nm and 2 nm and is produced upon BN adhesion on the metal surface. Correspondingly, 13 B or N atoms are located on 12 Rh atoms, which in turn determines the corrugation of the BN system. The periodic repetition of peaks and valleys originates in a uniform BN single layer with potential applications in spintronics, quantum computing, and photochemistry. It has been demonstrated that the nanomeshes could work as templates for molecules and metal nanoparticles. The molecules tend to be trapped into the pores preserving their structures and properties; this opens up new opportunities for molecular electronics [18].

h-BN represents a candidate for p-type layers in nitride deep-ultraviolet (DUV) emitters. Even though the lattice mismatch of h-BN with respect to the c-plane of AlN is more than 19%, five lattice constants of BN align with four lattice constants of AlN, reducing the mismatching down to 0.58% and making feasible a h-BN/w-AlN heterojunction. Moreover, h-BN is a good e-blocking and p-contact layer with w-AlN, showing a type II alignment [19].

Because of their negative electron affinity, BN nanotubes display interesting electron emission properties. Despite the higher turn-on field compared to carbon nanotubes (CNT), BN nanotubes have better thermal stability and can work at high temperature in an oxygen environment [20].

The great versatility of BN materials enables their application as neutron detectors [21] because of the large cross section of ^10^B for thermal neutrons. Maity et al. developed a h-BN detector, composed of two bilayers Ni/Au as ohmic contacts. They exhibited an efficiency of 58% when exposed to a calibrated neutron source [22].

A porous amorphous BN has been recently proposed as photocatalyst for the reduction of CO_2_. It works in the UV region without employing any cocatalyst [23]. Continuous endeavors to extend the BN application to the visible range have been made to trigger the reactions under solar exposition [24]. Several semiconductors in hybrid systems with BN were proposed such as TiO_2_, γ-C_3_N_4_, ZnO, In_2_S_3_ and many others [24]. In the composites, BN plays a crucial role, promoting the reduction of band gap, through structural strain and inhibiting electron hole recombination [25].

Since the sixties, BN has attracted considerable attention in aerospace applications [26]. Most recently, BN materials have been the focus of a renewed interest in the form of BN nanotubes (BNNTs), whose mechanical and thermal properties can be exploited for producing radiation shields, mainly as a reinforcing component [27].

Today’s BN research activity can be divided into two main areas. The first regards the improvement of mechanical and thermal properties. The obtained results are somewhat encouraging and involve both the main crystalline phases, h-BN and c-BN. The second focuses on the creation of nanosheets, controlling the number of layers and their structure with a particular interest in 0- dimensional systems (dots). BN-dots, in particular, have potential applications in several fields such as lighting, catalysis, and bioimaging.

In this brief review, we have considered the most significant results in the synthesis and processing of h-BN systems. We have analyzed the connection between dimensions, shape, and the growth process together with some of the accomplishments of theoretical calculations to identify the most probable defects in the hexagonal structure.

## 2. Understanding BN Materials by Theoretical Calculations

Despite the potential applications in several technological fields, 2D materials are still at their early stage. One of the main challenges is the possibility of interfacing two different layered materials to create new devices. This system would combine the electronic and optical functionalities from every single 2D material [28]. Computational investigations of electronic and optical boron nitride- based materials can provide a more in-depth theoretical insight, paving the way to a more efficient fabrication of devices. The possibility of nanoscale functionalization of the BN crystal structures is the basis of the great attractiveness of this material. In particular, quantum dots exhibit new interesting properties, directly linked to the size of the systems, passivation of their edges, shapes, and atomic or molecular doping [29].

The electronic structure that determines, in turn, the material property, can be computed by density functional theory (DFT)-based models. DFT can calculate and explore magnetic and optical properties, the system geometry and stability, as well as charge transport and structural transformation. Thanks to the increase of computational power, the systems that can be addressed can count thousands of atoms [28]. Although the accessibility to the DFT computing systems is growing, thanks to the increasing availability of user-friendly computing codes such as Quantum Espresso, Gaussian, Siesta, ABINIT, and VASP, the limitations and the efficiency of calculations are still under debate. Reliable correspondence of the calculations with experimental measures represents the main problem for providing a suitable complement to the experimental field [28]. The accuracy of the band gap calculation and the excited states investigation represent two of the main challenges for DFT applications. Standard DFT underestimates the band gap energy and needs to be corrected with the Green’s function quasiparticle (GW) method. This computing method has a quasi- particle approach scaling with N^4^ and, therefore, becomes very expensive for large systems. HSE06 is another choice frequently employed for van der Waals heterostructures. For the calculation of the excited states, time-dependent DFT is the most used, and its results are rather reliable [28].

Berseneva et al. [30] carried out a calculation of the h-BN electronic structure and studied the effect of carbon incorporation on the system properties. GW approximation showed that the band gap of 2D BN depended on the interlayer distance in the simulation supercell and it differed from bulk band gap by more than 0.5 eV [30]. Band gap values have been calculated using Perdew, Burke, and Ernzerhof (PBE) and Heyd–Scuseria–Ernzerhof functionals (HSE) approaches and GW approximation (with PBE data as input). They reported an indirect band gap of 4.56 and 5.56 for PBE and HSE, respectively. GW calculation seems to fail in the estimation of h-BN sheet, providing a succession of different energy band gap values for a wide range of interlayer distances (up to infinite separation) with no effective converged value [30]. The effect of carbon impurities has been tested in four different configurations by inserting a single C-atom in N or B sites and a triangular-shaped carbon unit. What has emerged from the defect formation energies is that a C atom tends to occupy the position of B atoms both in N-rich and B-rich environments. New states, close to the conduction(valence) band appear in the band gap with the substitution of B(N) atoms [30]. When a B(N) atom is substituted by a carbon atom, an electron (hole) is introduced into the structure, along with the formation of defect levels within the HOMO–LUMO gap of BN quantum dots (BNQDs) [31]. The substitution of B(N) by C-atom moves the Fermi level to a high(low) energy and the levels near the Fermi level are contributed by a C 2p_z_ orbital, as shown by the electron density of the state (PDOS) [31]. Moreover, Zhao at al. [31] found that in BNQDs, the C-atom tends to reside in the inner region of the minority sublattice. For the 2C-doped structure, instead, the C-atoms are preferentially located in the edge positions [31]. DFT calculations on B_11_C_12_N_9_ (doping with aromatic carbon) compounds have shown a remarkable reduction of band gap down to 2 eV, unveiling potential photocatalytic applications in the visible range [32]. Interestingly, Gao et al. [33] showed that C doping in the B position turns the BN sheets into a catalyzer for O_2_ activation. This effect is due to the electron transfer from carbon impurity to the 2π^*^ antibonding orbital of O_2_.

Spin-polarized calculations within GGA (generalized gradient approximantion) on electron-acceptor tetracyanoquinodimethane (TCNQ) and electron-donor tertrathiafulvalene (TTF) physisorbed on BNQDs have shown a low dopant-QD interaction due to the absence of a π-surface [29].

The lower the dimension of the system, the more the edge effects become relevant for structural stability and electronic and magnetic properties. Krepel at al. [34] demonstrated that the hydroxylation of zig-zag edges of BN is much stronger for nitrogen-rich edges and can produce considerable stabilization of the whole system. This is the most favorable scenario for particles growth in hydrothermal conditions or exposed to water vapor atmosphere. The results were obtained within the B3LYP hybrid exchange-correlation density functional approximation and show that hydroxylation reduces the band gap by more than 30%. This finding points out the importance of edge chemistry on the electronic properties of BN nanomaterials [34].

BNQDs could display innovative and interesting properties when functionalized with chemical ligands, especially for biomedical and optoelectronic applications [35] Hybrid DFT and Green’s function calculations have been carried out on h-BNQDs functionalized by -H, -OH, -SH, -NH_2_ and -N. As a general trend, the functional groups contribute to reducing the energy band gap. Moreover, the presence of side defects, working as single quantum emitters, could promote the lowest excitation energy in the visible and near infrared regions [35].

Defects engineering in BN nanomaterials is expected to open new frontiers in quantum technologies, such as quantum computing, communications, and metrology [36,37]. Single-photon emission from 2D BN was experimentally demonstrated by Tran et al. [36]. The bright emission, recorded at 623 nm at room temperature, was attributed to radiative recombination by structural defects. Attaccalite et al. [38] predicted a UV emission associated with boron vacancies (V_B_). DFT calculations with PBE exchange-correlation functional showed that the most likely candidate for the quantum emitter is N_B_V_N_, i.e. an anti-site nitrogen-vacancy (nitrogen occupies a boron site, and there is a vacancy at the nitrogen site) [36]. In the framework of quantum computing, group analysis along with DFT calculations demonstrated that the neutral paramagnetic carbon defect C_B_V_N_ could be a valid candidate for spin coherent manipulation and qubit, as well as C_N_V_B_ [39,40,41]. Recently, Weston et al. attributed the single-photon emission in the UV and at about 2 eV to carbon substituting nitrogen and interstitial centers, respectively [42]. According to the formation energy, it is more likely to be the incorporation of an oxygen atom in an N site rather than in a B site. O_N_ defect has been reported to lie at 5.3 eV, above the valence band, and it acts as a donor. V_B_O_N_ could introduce more levels inside the gap at lower energies, but it has high formation energy. Oxygen can be present in the form of interstitial defects (O_i_) at 2.93 eV above the valence band. Furthermore, thanks to their high mobility, even at room temperature, O_i_ can form complexes in place of isolated defects [42].

## 3. Making of 2D BN Materials

According to the available fabrication methods, we can distinguish the preparations of nanomaterials by two different approaches, namely the two typical top-down and bottom-up routes. Their applicability relies on the dimensions, purity, costs, and time necessary to obtain the final products.

The 2-D materials represent a peculiar case, where the possibility to unleash the outstanding properties deriving from their layered structures, represents the primary goal of every growth approach. The choice of specific synthesis and processes depends on several key factors such as yield, ease of implementation, production pace, defects generation, precursors, and potential scalability [43,44]. The production of monolayers or a few layers of h-BN is the main target of the two methods. With a very rough simplification, the top-down approach is based on the exfoliation of the stacked structure of raw BN materials (mainly in the form of powder). The exfoliation, in general, produces BN sheets of a few layers. The chemical reactions of molecular precursors are at the base of the bottom-up approach [45,46]. Liquid phase methods are also employed for the fabrication of nanosystems. Thanks to the reduced processing cost, they are promising routes for large scale production. Regarding the case of BN, hydro/solvothermal methods are suitable for the preparation of dots, small clusters, and flakes [47,48,49]. Besides, it is not unusual to carry out a combination of top-down and liquid phase synthesis with the purpose of enhancing the reduction of bulk BN in sheets and functionalizing them [50,51,52].

### 3.1. Top-Down Synthesis Routes

When top-down synthesis is employed, a large piece of material (bulk) transforms into a fine powder by applying a certain amount of external energy until the system is scaled down to nanosize. Among others, mechanosynthesis is a low-cost and reliable method for nanomaterials production [46]. The main result of the mechanical approach is the exfoliation into layers of various sizes, which can be made smaller by further processing cycles.

Since its application to graphene [53], the “scotch-tape” method has been considered easy and reliable for obtaining a small amount of 2D materials. The weak Van der Waals forces can be overcome by subsequently tear movements until the monolayer is reached [54]. As shown by Tang et al, the peeling-off process bends and extracts the layer preserving its high crystallinity and related electronic properties [55]. Despite the advantage of high-quality samples, BN cleavage in sheets by the scotch method does not enable large scale uses but only laboratory applications [10].

Another top-down method is ball-milling. A ball-milling system provides the kinetic energy for reducing the material into small fragments. Besides, the involved energy may start chemical reactions, paving the way for fast organic synthesis [56]. Many parameters control the ball-milling processes [57,58] such as ball-to-power ratio, processing time, milling speed, and ball size. The choice of the parameters depends on the desired dimensions and, in the case of BN materials, also affects the exfoliation process. Sheets of BN have been produced by applying shear forces to break the interplanar bonds. A milling speed of 800 rpm with a ball-to-powder weight ratio of 10:1 is capable of producing exfoliated BN sheets efficiently. With 10 h of treatment, 0.1–0.2 mm balls could provide crystalline BN sheets with low damaged structures. The ball-milled products were then tested with oil and showed an improvement in lubricating properties [57] (Figure 2).

The peeling mechanism for BN sheets is solvent dependent. Li et al. [59] demonstrated the effectiveness of benzyl benzoate over other milling agents like water, ethanol or dodecane. The benzyl benzoate enhances the exfoliation process due to its higher viscosity and hinders the formation of sheets agglomerates. Besides, due to the low reactivity of benzyl benzoate with Fe and the higher viscosity, which reduces the ball impact, the authors measured less steel contamination (from the vials and balls) in comparison to other milling agents [59].

High-pressure microfluidization has been used for cleavage BN micropowders into thin fragments with encouraging results [60]. A solution of BN precursors in DMF/chloroform was accelerated by a pump up to 400 m s^−1^, with a pressure of 2069 bar, before being introduced into an interaction chamber made of multiple microchannels. The BN precursors were forced to collide at high energy, breaking into small pieces, approximately of few microns in-plane and 12 nm in thickness, corresponding to 20–30 BN monolayers (Figure 3) [60].

A less sophisticated apparatus based on a rotating glass inclined at 45°, was used to combine centrifugal and gravitational forces to enhance the exfoliation of graphene and BN in N-Methyl-2-pyrrolidone (NMP). BN sheets with an average flake dimension of 2–3 µm were obtained using an optimal speed of 8000 rpm [61].

Despite the method being simple and the quality of the nanosheets being good [59], ball milling and vortex fluidic methods turn out to be still inefficient for the preparation of 2D BN in high yields. This low efficiency hampers its applicability for large scale manufacturing [62].

The exfoliation process with the highest yield is chemical assisted exfoliation [62,63]. With the aid of a sonication bath, the layered BN structure can be reduced in foil and dispersed in a solvent. The efficacy of the process depends on the type of solvent, energy, and time [64,65]. The sonication treatment generates a succession of compression and rarefaction. The cavitation microbubbles tend to grow at each cycle, collapse, and generate shockwaves [65,66].

BN exfoliation was reported for the first time by Han et al. An amount of 0.2 mg of BN single crystals was added to 5 mL of a 1,2-dichloroethane solution of poly(m-phenylenevinylene)-co-2,5-dioctoxy-p-phenylenevinylene for 1 h. The sonication process was able to break up the BN bulk into mono- and a few layers. The time treatment represents a compromise between sonication efficacy and damage of the structure [67].

The correct choice of the solvent is important to consider for the surface energy of the nanosheets. The most efficient solvent is one with a similar surface energy. This worked very well with isopropyl alcohol (IPA) and N-methyl-2-pyrrolidone (NMP) for BN, which possesses a surface energy of 65 mJ m^−2^ [45,68,69]. Since then, many other solvents have been tested such as dimethylformamide (DMF) [70], ammonium hydrogen carbonate [71], methanol [72], sulfuric acid [73] or ammonia [74].

Water represents a particular case of BN exfoliation. Li et al. have demonstrated the high effectiveness of water in the production of nanosheets. However, BN tended to react with water forming ammonia and underwent edge functionalization with OH groups [75].

BN edges can be affected by protonation in the presence of acid. The presence of repulsive charges can favor the separation of layers, as in the case of graphite [76]. The typical yield of dispersed BN nanosheets, in a mixture of H_3_PO_4_ and H_2_SO_4_ (ratio 1:8), is between 12% and 15% of the precursors [76], comparable with methanesulfonic acid (CH_3_SO_3_H) [77].

In summary, the progressive reduction of the bulk material down to nanometric sizes is the basis of the main top-down methods. For a 2D material, this implies progressive exfoliation of the system into one or a few layers. We have seen how, to date, top-down processes suffer from poor material yield, but have the advantage of affecting the crystalline structure very little, except at its edges, where functionalizing molecular groups can be easily present. For example, in an OH-rich environment, the presence of such groups in BN nanostructures is common.

### 3.2. Bottom-Up Synthesis Routes

In the bottom-up approach, the final composite material is prepared by starting from the elemental or molecular components and allowing them to react and assemble into the required structure.

One of the main techniques which guarantees a high purity and low defect layered material is the chemical vapor deposition (CVD) method. The first work on hexagonal BN was reported by Paffett et al. [78]. They described the procedure for a single layer deposition of BN on Pt (111) and Ru (001) substrates. The data collected on adsorption and decomposition of borazine (B_3_N_4_H_6_), revealed a better reactivity for Pt substrate with a coverage ratio of 0.36 against 0.15 for Ru. Later, Nagashima et al. [79] successfully deposited a layer of h-BN on a substrate of Ni (111) heated at 800 °C.

Both of the experiments mentioned above used borazine as the precursor. Since borazine is toxic and liquid at room temperature, considerable efforts have been made to find alternative precursors for h-BN by CVD. Among others, B-trichloroborazine (ClBNH)_3_ shows stability and easy handling [80].

The CVD method is affected, in general, by the precursor chemistry and substrate. Corso et al. unveiled the impact of the substrate on h-BN morphology, showing how the nanomesh of BN resulted on Ru (0001) and Rh (001), while a flat layer was obtained on Pd (111), Pt (111), Cu (111), Mo (110), Cr (110), Fe (110), and Ni (111) [15]. If polycrystalline substrates of Ni or Cu are employed, the CVD method can also provide high purity BN nanosheets with decaborane/ammonia as precursors [81]. Although CVD has the advantage of being able to grow single BN sheets of high quality on a large variety of substrates, with high control of possible foreign elements, this technique is unfortunately still far from being used for large regions of the substrate and for high control over multiple layers. In this respect, Jeong et al. displayed encouraging results (Figure 4). They grew large scale h-BN on Ni (111) substrate; the ammonia, used as a precursor, tended to be adsorbed and decomposed on Ni. The decomposition produces radicals and forms an extended portion of high-crystalline h-BN. This effect turned out to be hardly reproducible on sapphire, demonstrating the importance of the catalytic effect of the substrate [82].

Pulsed laser deposition is a promising technique for the growth of extended films on different types of substrates. The advantage of this method is the possibility of covering a surface, regardless of the substrate. This property allows the growth of BN films on different types of materials, according to the specific applications. Acacia et al. [83] investigated the role of substrate temperature on BN quality, keeping the laser fluence constant at 1.35 J cm^−2^. The experiment was carried out in a high vacuum chamber (<10^−4^ Pa) by exposing a rotating BN target to a KrF laser irradiation and using <111> c-Si and Corning glass as substrates. Two substrate temperatures, mainly at 25 and 600 °C, were analyzed [83]. The content of nitrogen at 600 °C decreased, and an excess of the metallic boron phase appeared. Moreover, the films showed both the presence of c-BN and h-BN phases in a largely disordered structure [83].

With the perspective of substituting Ag single crystals with less expensive Ag thin film as a suitable substrate for BN deposition, Velazquez et al. [84] deposited good quality h-BN single layers on Ag(111)/SrTiO_3_(001) substrates (Figure 5).

The stoichiometric BN films were grown by using the fourth harmonic of an Nd:YAG laser with a fluence between 5 and 6 mJ cm^−2^ and presented a sub-millimeter extension with several flakes on it [84].

A proper choice of the substrate can easily tailor the phase of BN films. Large and stoichiometric h-BN films have been grown by pulsed layer deposition (PLD) on highly ordered pyrolytic graphite (HOPG). The layered structure consisted of 4 to 6 planes after 5 s of deposition on 25 mm^2^ at 700 °C [85]. Besides, amorphous BN was successfully grown on sapphire (0001). Both crystalline and amorphous materials had high quality and low defects. This route may open the way for film deposition on large surfaces for commercial purposes [85].

Magnetron sputtering is another common approach for physical vapor deposition of films with advantages of simplicity and cost-effectiveness. The method employs a magnetic field to confine the electrons in the region of the sputtering target. A gas (typically N_2_ or Ar) is ionized by the electron cloud and accelerated towards the cathode. The transfer of momentum ejects the particles from the target, condensing them on the substrate. This technique is considered particularly reliable for thick c-BN and for BCN coatings on cutting tools with high resistance surfaces [86].

Molecular beam epitaxy (MBE) promises to be a valid unconstrained method for high scalable methods, in contrast to CVD which is based on the decomposition of molecular precursors and its strong interaction with the substrates [87]. One of the pioneering works is from Tsai at al. They demonstrated the reliability of plasma assisted MBE methods for the growth of thin BN films on Ni (111). Ni substrate was preheated at 400 °C for 20 min for preliminary cleaning before transfer into an MBE growth machine and then heated up to 900 °C to remove native oxides and preparation for exposure. High purity boron was evaporated with an electron beam gun at an evaporation rate of 0.9 A/h; N was provided by a high purity N_2_ gas flux. The h-BN was grown at a rate of 650 A/h and a temperature of 890 °C. Reflection high-energy electron diffraction (RHEED) patterns and X-ray diffraction (XRD) have shown that BN films of good quality can be produced [88]. Polycrystalline Ni has also been used for MBE growth of BN thin films. An effusion cell at 1850 °C for elemental B and RF plasma source at 350 W for N was used as a precursor [89]. Raman spectroscopy measurements confirmed the high crystalline degree of BN films of hexagonal phase, which exhibit a characteristic sharp peak at 1365 cm^−1^ (*G* band corresponding to the E_2g_ vibration mode). Despite the homogenous covering of the Ni substrate, atomic force microscopy (AFM) images revealed the presence of ridges with two different features, a wrinkled structure and discontinuous dotted lines. The former defects are likely related to the discrepancy between the BN and Ni thermal expansion coefficient but the origin of the latter is still unknown. The process of MBE deposition on poly-Ni started with nucleation of small ramified islands at 730 °C which changed shape into triangles when the temperature was increased up to 835 °C. The nucleating domains coalesced and evolved into compact films [89].

One of the main advantages of the MBE technique is the possibility to efficiently control the deposition of thin films layer by layer on the atomic scale, without decomposition of the molecular precursors but by assembling the elemental precursors. This deposition control is favorable for 2D materials, where other techniques such as CVD face the problem of depositing more than a single layer due to the van der Waals forces.

Tonkikh et al. demonstrated the feasibility of achieving precise control of A-B stacking of BN layers. According to the procedure, the Ni (111) substrate was thermally treated at 800 °C. Then followed the deposition of a Ni buffer layer by e-beam evaporation and a single layer of B. Up to 10 high-quality layers were piled up [90] (Figure 6).

### 3.3. Synthesis Route for BNQD

The increasing interest in 2-dimensional materials of a few nanometers in size has turned attention onto bottom-up methods from solutions. These methods, being relatively simple and cost- effective, allow the production of 0-D nanomaterials, known as quantum dots. 0-D BN materials (BN dots) are expected to have several high impact applications in photonics, (photo)catalysis, and biomedicine [91]. They have an intrinsic advantage of a high surface/volume ratio.

Most of the mentioned synthesis methods represent valid solutions for the realization of a single layer on substrates, like CVD, or of a few layers through ultrasonication. However, many of the most relevant properties of BNs are on the nanoscale (on a few nanometers), where the control of the edges and defects become of primary importance. CVD or sonication fails in further reduction of the h-BN monolayer dimensions, preventing access to the properties of BN-based QDs. New specific techniques for the control of size and structure have been developed for this purpose.

Lin et al. reported a method for the fabrication of monolayered BNQDs by exfoliation and disintegration of h-BN flakes (Figure 7). BNQDs of 10 nm and a direct bandgap of 6.51 eV, attributed to the quantum confinement effect, were obtained. The monolayered h-BN presented a blue-green luminescence excitable in the UV region (365 nm). The authors assigned the QDs’ emission to three types of optically active centers: N vacancies replaced by C atoms (λ_em_ = 423 nm), carbene structures saturating the zig-zag edges (λ_em_ = 420 nm) and BO_x_^−^ (x = 1,2) species (λ_em_ = 425 nm). The h-BNQDs quantum yield (QY) is 2.5% [92].

Stengl et al. observed similar PL features for samples realized by a high-intensity cavitation procedure. Bulk h-BN was suspended in water and exposed to high intensity ultrasound in a reactor at a pressure of 6 bar for 5 min. The layers were then refluxed in ethylene glycol at 198 °C under atmospheric pressure for 48 h [93]. The dots showed a few layers’ composition (less than 5 nm). The h-BN dots had a broad and asymmetric emission peaked at about 470 nm upon excitation at 365 nm. At least two components are the source of the emission [93]. Although the authors did not suggest any attribution, the PL appears to be excitation dependent with presumable excitation selectivity of the defects.

Following a similar approach, Li et al. synthesized BN nanosheets by a sonication treatment of BN powders in dimethylformamide (DMF) for 8 h. After the removal of larger particles by centrifugation, the nanosheets were solvothermally treated in a Teflon-lined autoclave at 200 °C for 24 h. This method allowed a yield of 21.6% to be achieved. HRTEM investigations revealed a good crystallinity of BNQDs with a lattice parameter of 0.21 nm. The dimensions of BNQDs depended on the solvothermal treatment time: 10.06, 4.12, 2.41 nm corresponded to the duration time of 6, 12, and 36 h respectively [94]. FTIR spectra revealed a strong absorption at 1378 and 800 cm^−1^ attributed to B–N stretching and B–N bending modes, respectively. The absorption bands at 2931 and 2857 cm^−1^ were then attributed to the CH_3_ group of the DMF, demonstrating the presence of solvent residuals on the QDs’ surfaces. Besides, the FTIR spectra showed the presence C–(BN), B–O and O–B–O vibrations at 1750–1550, 1150–850, and 750–450 cm^−1^ and absorption bands at 1386–1450 cm^−1^, caused by edge oxidation [94]. The PL QY is around 19.5%, a significant improvement compared to previous works as for Lin et al., the PL spectra had an excitation energy dependence. Again, the authors attributed the dominant emission at 395.5 nm (excited at 375 nm) to BO_2_^−^ centers, with other contributions from carbene edge and 3-B centers [94].

Analogous results were found by Jung et al., who forced the presence of defects on h-BNQDs by striking them with iron nanoparticles and treatment under microwave-sonication in water. As shown by FTIR spectra, the treatment causes hydroxylation of the BNQDs edges as demonstrated by an increase of –OH stretching mode intensity. XPS spectra exhibited the B 1s and N 1s peaks along with C 1s and O 1s in both prepared BN and hydroxylated BN samples, demonstrating the presence of a carbon and oxygen atom in the structure of the BNQDs [95]. PL spectra have two emissions at 320 nm (excitable at 280 nm) and at 450 nm (with excitation at 280 nm and 360 nm). The authors attributed the UV emission to the presence of nitrogen and boron nitride, already seen in bulk h-BN. The emission in the visible range was ascribed to oxygen impurities and decreased in the presence of hydroxylated groups [95,96]. Moreover, the PL emission had a pH-dependent modulation, which is typical of defects sites of oxygen on 2D materials [95].

BNQDs grown by a hydrothermal route in water using boron acid and ammonia as precursors showed a visible emission centered at 400 nm. This emission was attributed to a charge-transfer mechanism deriving from the hydroxyl groups, which receive in turn electrons from core nitrogen atoms under UV excitation [47]. Solvothermal synthesis starting with BN nanosheets, revealed that the blue emission is dependent on the solvent polarity [97]. The higher the polarity of the solvent, the more significant is the redshift. The largest redshift was measured in smaller BNQDs (about 2 nm in size) which excluded any quantum confinement effect. On the contrary, the PL shift should be connected to chemical species or defects in the QDs structures. As reported by Liu et al., the content of oxygen resulting from the higher polarity solvent (NMP in this case) is responsible for emission at larger wavelengths [97].

The origin of BNQDs luminescence has been supposed to originate from the presence of chemical species attached to the surface. The bottom-up synthesis methods, employing an organic solvent or carbon-based precursors can promote the functionalization of BN nanosheets, which in turn display new emissions in the visible range.

The –OH groups at the surfaces of BNQDs, grown by hydrothermal methods, can be exploited to selectively detect metal ions for biological labelling or fluorescent probes purposes. Huo at al. synthesized OH rich QDs by a hydrothermal route [98]. As shown in Figure 8, upon the addition of Fe^3+^ ions, the BNQDs underwent a strong fluorescence quenching, under UV excitation. This effect was not reproduced with other ions, such as Al^3+^, Cd^2+^, Cu^2+^, Co^2+^, Pb^2+^, Fe^2+^, Mn^2+^, Ba^2+^, Ni^2+^, Hg^2+^, and Ag^+^. The authors attributed the phenomenon to the strong binding affinity of electron-deficient Fe^3+^ to the electron-rich hydroxyl groups [98].

Most of the BNQDs exhibit a low quantum yield, which is highly dependent on the synthesis process. Lei at al. achieved an absolute QY of 8.6% with their BNQDs realized in DMF and DMSO (dimethyl sulfoxide) after exfoliation [51]. Fan et al. demonstrated the efficacy of microwave irradiation over solvothermal methods, in terms of energy and time-saving [99]. The nanoplates of BN obtained by sonication in DMF were irradiated under microwave at a power of 500 W at 150 °C for 10 min. The BNQDs, with an average size of 1.98 nm, had a strong emission peaked at 426 nm and QY = 23.44%, measured using quinine hemisulfate monohydrate as the reference [99].

A QY of 32% was measured for blue-emitting h-BNQDs, synthesized by a hydrothermal process by Dehghani et al. [100]. After further surface passivation using PEG_200_, the h-BNQDs displayed an increase of QY up to 38% (again measured by using quinine sulfate as reference) and a slight redshift. The effect of surface passivation was analyzed by FTIR and XPS measurements. Pristine and functionalized samples presented similar chemical species, corresponding to B-O, B-N and oxygen rich functional groups (derived from the exfoliation process). The functionalization process seemed to alter the surface chemical composition favoring the formation of nitrile groups, thereby causing a luminescence redshift [100].

More recently, BNQDs were successfully produced by a hydrothermal route, employing boric acid and ammonia (H_3_BO_3_:NH_3_ = 1:6) at 200 °C for 10 h [101]. The dots, with an average dimension of about 10 nm, did not display a pure B–N structure. On the contrary, FTIR, XPS spectra, and XRD patterns show a clear presence of B–O, B–N–O and N–H bonds and a non-negligible percentage (around 5.5%) of unreacted boric acid. Further thermal treatments up to 300 °C increase the contribution of the B-O component as an effect of the oxidation process. The obtained nanoparticles presented a boron-oxynitride (BOND) network without carbon as contaminant agent [101]. UV-Vis spectra of the BOND particles had four contributions at 220, 265, 308, and 412 nm. The energy band gap was measured at 5.03 eV, lower than the 5.81 eV of BN sheets [102], which increased under high temperature treatment due to the reduction of OH groups. The bands at 220 and 265 nm are not optically active, whereas the excitations at 308 and 412 nm produce an emission between 350 nm and 600 nm. 3D PL excitation-emission-intensity spectra revealed the dual nature (two-color emission) of BOND luminescence, with two maxima at 390 nm and 470 nm (Figure 9) [101].

Table 1 lists a summary of the most relevant results. The absolute efficiency of BNQDs is generally below 10%. The higher QYs are measured for some BNQDs produced by employing precursors bearing carbon atoms, which end up doping the BN structure. Furthermore, many of them report a QY obtained through the use of a standard as reference rather than an absolute integrating measurement [94,99,100,103,104] which could in principle provide an overestimation of the luminescence yield [105]. Carbon and oxygen doping inevitably influences the emission mechanisms and their spectral characteristics, according to modalities not yet investigated.

## 4. Summary

What emerges from this review is that boron nitride-based systems are still far from having been comprehensively studied and their properties unveiled. BN in its 2-D and 0-D forms is much less understood in comparison to graphene. The BN system turns out to be an extremely stable material, both chemically and thermally. It presents interesting applicative perspectives in electronics as an insulator on a nanometric scale and can be easily interfaced with its conductive counterpart, graphene. Up to now, the possibilities of practical applications extend to very varied fields, from mechanics to optics. However, from careful observation of the experimental results, the exploitation of the mechanical and thermal properties is much more convincing. It is worth emphasizing, once again that the recent development of materials based on BN nanotubes for thermal applications are not addressed in this brief review.

The deposition processes of single or few layers on large regions remain at the level of basic experiments. Although some methods are progressing in terms of material quality and extensions, the most favorable substrates and the choice of the most common precursors are still under study.

Furthermore, defect control seems to be far from being fully understood. We have seen how researchers apply an approach which is like that of carbon-based materials. The target is to obtain 0- dimensional systems as visible emitters for uses in lighting or diagnostics. It appears, however, virtually impossible to obtain a homogeneous structure of BN, without foreign atoms which partially modify the structure. The top-down methods allow, in principle, to derive BN nanosheets by ultrasonication. It is, however, very complicated to be able to control the exfoliation process down to the monolayer and the process yield is still rather poor for scaling up. Bottom-up solution systems have given the most encouraging results in terms of optical emissions. However, the emissions are due to defects or foreign atoms introduced during the process which originate from precursors and solvents. Hydrothermal routes, on the other hand, contribute to the functionalization of the BN system with OH groups. Even if the functionalizing groups or the doping atoms contribute to the reduction of the optical band gap and the emission in the visible range, the doping mechanisms are not yet clear. The theoretical studies highlight the nature of the most probable BN defects which are vacancies of B or N, interstitial defects, or chemical groups derived from interaction with the growth environment, as in the case of BOs.

However, until now, no systematic study aimed at labelling each of the BNQDs emissions has been carried out. Many of the experimental works are limited to reporting multiple recipes for the growth of BNQDs, measuring visible emissions without an in-depth study of the growth process impact on the final performance of the BN.

Most of the works on BN are devoted to realizing dots of high emission in the visible and with high quantum yields. Unfortunately, a description of BN dot radiative recombinations is still lacking. It is also not clear what the effect is of the substitution or elimination of some atoms.

We believe that it is essential to perform specific studies to assess the origin of the emission properties of fluorescent dots. In a scenario where research is still in its early stages, this would allow full exploitation of the potential of BN-based low-dimensional materials.

## Figures and Tables

**Figure 1 materials-12-03905-f001:**
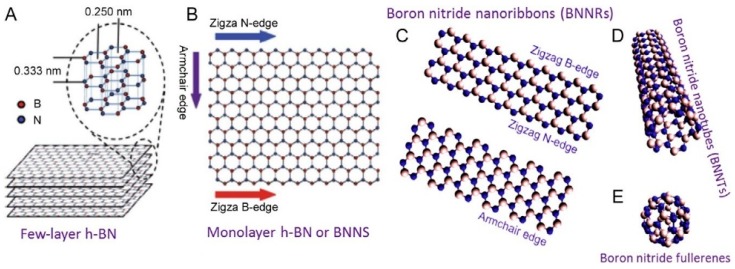
(**A**) hexagonal boron nitride structures. Layer distance and crystal parameter. (**B**,**C**) Different edge terminations. (**D**,**E**) different boron nitride shapes: BN nanotubes and BN fullerene. Copyright 2012 and 2014, with permission of refs, [7,8].

**Figure 2 materials-12-03905-f002:**
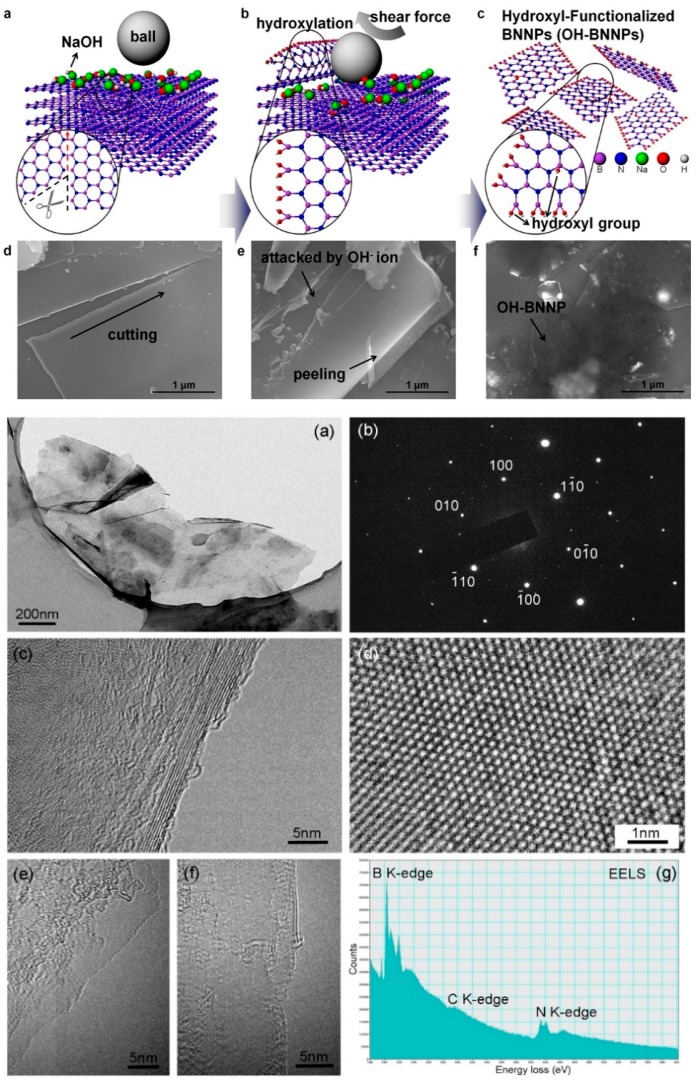
On top: Schematic diagram of exfoliation mechanism. Copyright 2015, with permission of [58]. On bottom: (**a**) TEM Images of the BN sheets produced by 0.1–0.2 mm balls; (**b**) the corresponding SAED pattern; (**c**,**d**) high-magnification TEM images; (**e**,**f**) TEM images of few-layer BN nanosheets (**g**) EELS spectra of the BN nanosheet. Copyright 2014, with permission of [57].

**Figure 3 materials-12-03905-f003:**
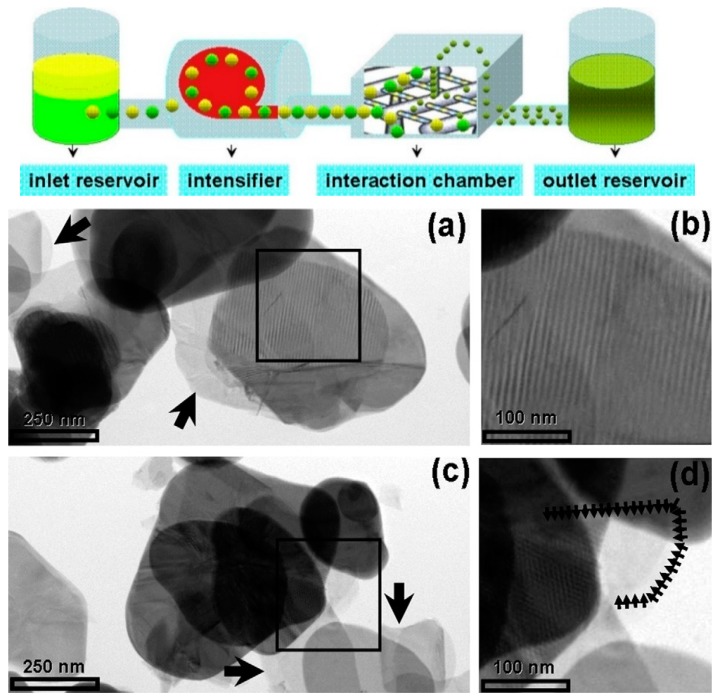
On the top: Schematic representation of a microfluidizer processor. Below: Low magnification TEM images: transparent regions indicated by arrows refer to a few layers of boron nitride nanosheets. Copyright 2012, with permission of [60].

**Figure 4 materials-12-03905-f004:**
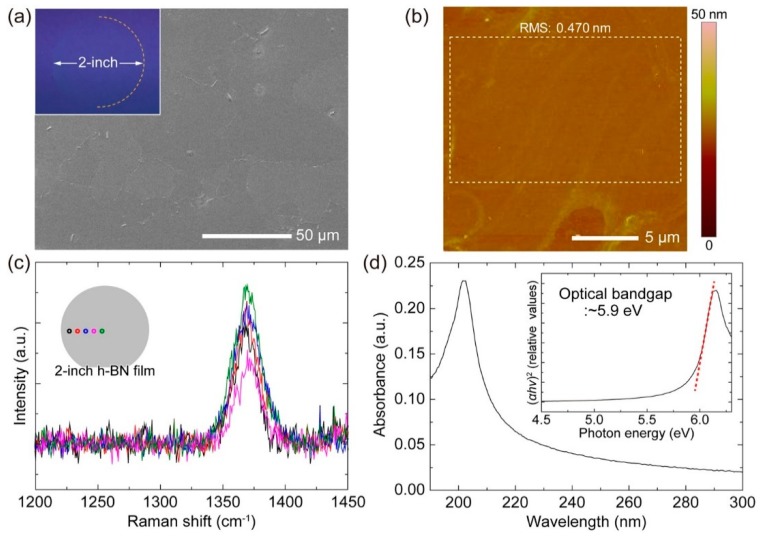
On top: CVD grown h-BN film on Ni (111). (**a**) SEM, (**b**) AFM, (**c**) Raman, (**d**) UV–Vis characterizations. On bottom: HR-TEM characterization of h-BN grown on Ni (111) (**a**) and sapphire (**c**). Cross-sectional HR-TEM of h-BN on Ni (111) (**b**) and sapphire (**d**). Copyright 2019, with permission of [82].

**Figure 5 materials-12-03905-f005:**
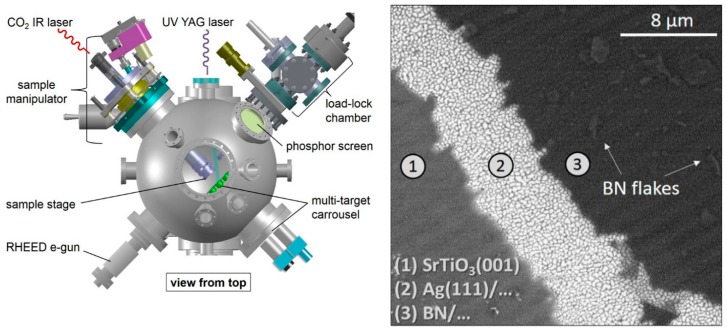
Left: scheme of PLD system. Right: SEM image of h-BN/Ag(111)/SrTiO3(001) deposited by PLD. Copyright 2016, with permission of [84].

**Figure 6 materials-12-03905-f006:**
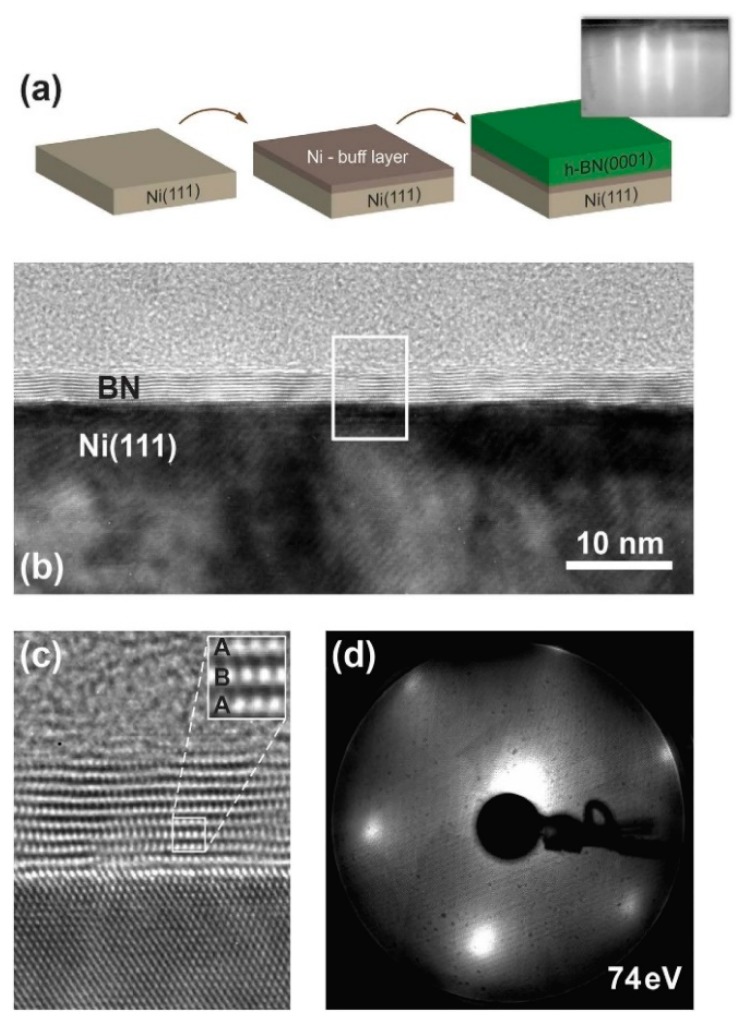
Growth of h-BN on Ni (111) by MBE. (**a**) deposition of oriented h-BN layer on Ni layer. (**b**) TEM image of multiple layers. (**c**) particulars of the layer stacking. (**d**) LEED image with electron beam energy of 74 eV. Copyright 2016, with permission of [90].

**Figure 7 materials-12-03905-f007:**
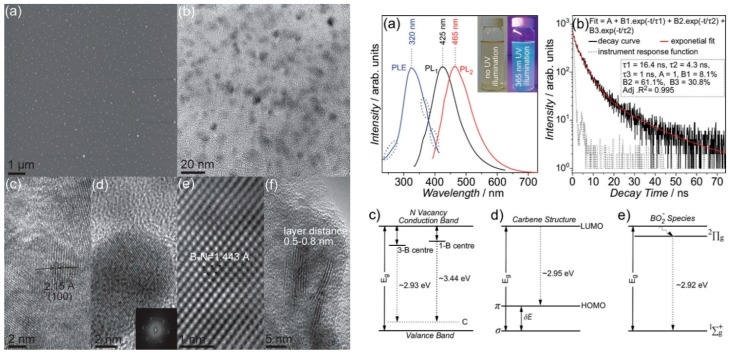
Left: AFM topography image of BNQDs. (**b**–**f**) HRTEM images of the BN dots. Right: PLE, PL and time-resolved PL spectra of BNQDs. (**c**–**e**) Energy states’ attribution to BNQDs luminescence. Copyright 2014, with permission of [92].

**Figure 8 materials-12-03905-f008:**
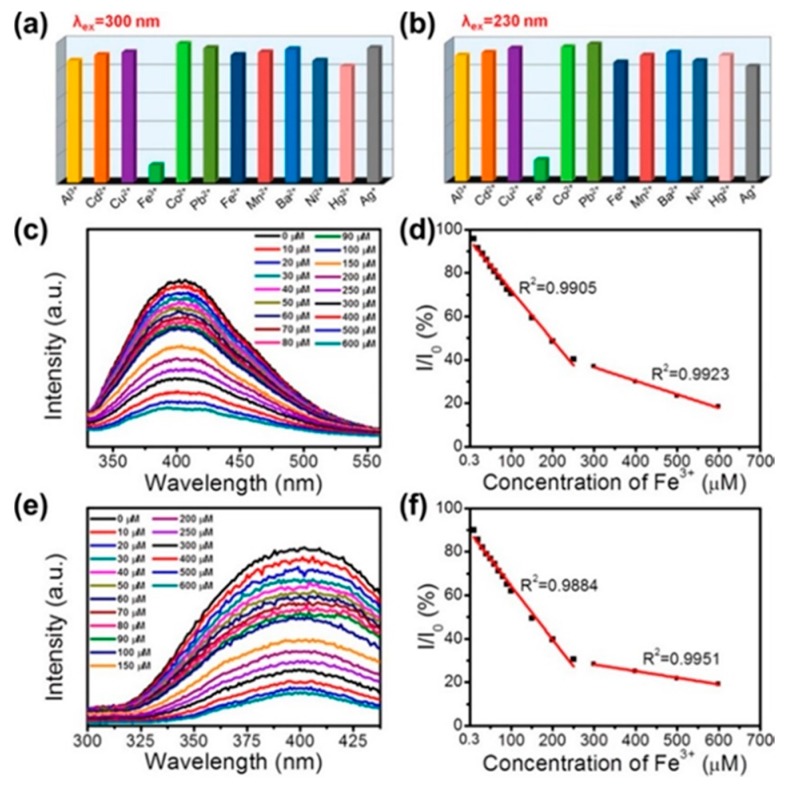
Hydrothermal synthesis route of BNQDs with boric acid and melamine. PL response of BNQDs measured by addition of different metal ions with a concentration of 200 µM at the excitation wavelength of 300 and 230 nm, respectively. Copyright 2017, with permission of ref. [98].

**Figure 9 materials-12-03905-f009:**
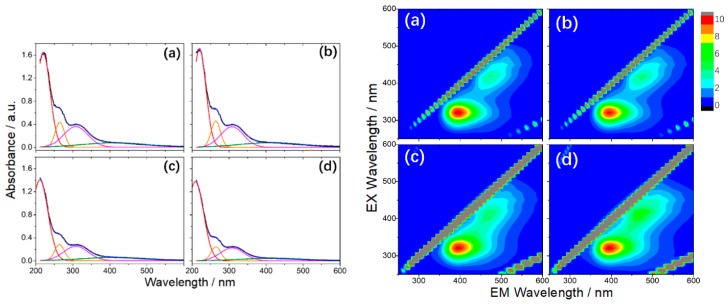
On the left: The UV-Vis absorption spectra and corresponding Gaussian fitting from (**a**) as prepared, (**b**) 100 °C, (**c**) 200 °C, and (**d**) 300 °C BONDs (in water, 0.1 mg mL^−1^). On the right: 3D PL excitation-emission-intensity spectra of (**a**) BONDs, (**b**) 100 °C, (**c**) 200 °C, and (**d**) 300 °C (in water, 0.1 mg mL^−1^). Copyright 2019, with permission of ref. [101].

**Table 1 materials-12-03905-t001:** Dependence of BNQDs structural and optical properties on the growth process. (**a**) boron source; (**b**) nitrogen source; (**c**) carbon source.

Precursors	Growth Conditions	Structure/Quantum Yield	Ref
(a) NaBH_4_ (b) CO(NH_2_)_2_ (c) CH_5_N_3_, HCl	Ball milling in a eutectic mixture of LiCl/KCl and then treated at 700 °C for 2 h	C, N-doped h-BNQDs (BCNO) nanoparticles with tunable luminescence from blue to green. a/b/c = 1/10/0, 1/0/1, 1/0/15, 1/0/10, 1/0/15 QY of 23%, 16%, 26%, 11%, and 5.6%	[103]
(a) B(OH)_3_ (b) CO(NH_2_)_2_ (c) PEG	Calcination at 750 °C for 1 h and ultrasonic treatment at 40 kHz for 3 h	BCNO QDs with a mean size of 10.1 nm. Broad luminescence with two components at 400 nm and 500 nm. QY = 19.9%	[104]
(a) H_3_BO_3_ (b) CO(NH_2_)_2_ (c) PEG M_W_ = 20000)	Calcination of the raw materials mixture at 800 °C for 30 min	BCNO with tunable luminescence from blue to red with atomic ratio B/N = 0.2, 0.3, 0.4, 0.5; B/C = 0.5. QY = 9.3%, 20%, 17%, 5.2%	[106]
(a) H_3_BO_3_ (b) CO(NH_2_)_2_ (c) C_6_H_8_O_7_	Microwave treatment at 800 W for 40 s after solvent evaporation	BCNO dots with tunable luminescence from blue to red. Molar ratio a/b/c = 1/1/2, 1/1/1, 1/2/1, 1/2/2, 1/1/0.5, 1/0.5/1, 1/0/1. QY = 50.9%, 40.0%, 7.5%, 19.9%, 23.2%, 41.2%, 3.5%	[107]
(a) B(OH)_3_ (b) CO(NH_2_)_2_ (c) C_6_H_12_O_6_	Microwave treatment at 800 W for 10 min	BCNO particles of 2 nm in size Blue emission peaked at 450 nm with QY = 27.1%	[108]
B(OH)_3_	Annealing at 800 °C for 5 h and hydrothermal treatment with NH_3_ (pH = 10) at 100 °C for 6 h and sonication for 2 h	BCNO nanoparticles with average size of 4.0 nm. PL emission at 410 nm with QY = 5%	[109]
(a) B(OH)_3_ (b) CO(NH_2_)_2_ (c) PEG (M_W_ = 20000)	Calcination at 700−900 °C for 30−60 min	BCNO (t-BN) nanoparticles of 5 nm in size. Tunable luminescence in correspondence of different c/a ratios. External QY = 79% (λ_em_ = 469 nm); 76% (λ_em_ = 520 nm); 53% (λ_em_ = 542 nm); 10% (λ_em_ = 571 nm)	[110]
BN powder	Sonication of BN powder in DMF for 8 h and solvothermal treatment at 200 °C for 8 h	BCN nanoparticles lower than 4 nm PL emission peaked at 396 nm, tunable down to 450 nm. QY = 19.5% (λ_ex_ = 330 nm)	[94]
BN powder	Sonication of BN powder in DMSO or DMF for 8 h	BNO nanoparticles of 2.5 nm. Blue PL emission peaked at 442 nm. QY = 8.6% (λ_ex_ = 360 nm)	[51]
BN powder	Sonication of BN powder in EtOH, DMF, NMP for 3 h and solvothermal treatment at 180 °C for 10 h	BNCO dots. Blue-green emission EtOH: 4 nm; QY = 12.6% DMF: 2.8 nm; QY = 16.4% NMP: 2 nm QY = 21.3%	[97]

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
