# Peer review of "From 2-D to 0-D Boron Nitride Materials, The Next Challenge"

_materials, 2019, doi:10.3390/ma12233905_

Round 1

Reviewer 1 Report

Too many topics are covered in too little depth throughout the paper. The author is correct in that there is not a developed technique to produce crystalline BN at any scale or dimensionality. The manuscript does provide a reasonable survey of possible applications for BN and most techniques used to synthesize BN, but does not provide any additional insight. Extensive editing of English language and style is required.

Author Response

Too many topics are covered in too little depth throughout the paper. The author is correct in that there is not a developed technique to produce crystalline BN at any scale or dimensionality. The manuscript does provide a reasonable survey of possible applications for BN and most techniques used to synthesize BN, but does not provide any additional insight. Extensive editing of English language and style is required.

Response: We are aware that every subject covered in this work should deserve specific attention and should be described in a much more extensive review. However, we wanted to draw the reader's attention to a detailed state of the art on boron nitride, highlighting the main drawbacks and problems of this material. We want to underline that the present work is a review, and no new insights are expected to be presented but rather a general overview of state of the art. The article, therefore, would represent the first step for future developments. As suggested, the article has been updated and corrected in style and editing.

Reviewer 2 Report

This manuscript presents a nice summary of boron nitride material- it's synthesis and related structural properteis. Author summarizes the process of realizing 2D BN as well as 0D BN and discuss the advantages and disadvantages of different techniques. It include different process for top-down approch and at the same time author discuss different technique of bottom-up approch including PLD and MBE.

the manuscript also has a useful summary of dependence of BNQDs structural and optical properties on the growth process.

There are many things which can be added to this review article, however, considering it's length, it can be accepted in present form.

Author Response

This manuscript presents a nice summary of boron nitride material- it's synthesis and related structural properteis. Author summarizes the process of realizing 2D BN as well as 0D BN and discuss the advantages and disadvantages of different techniques. It include different process for top-down approch and at the same time author discuss different technique of bottom-up approch including PLD and MBE.

The manuscript also has a useful summary of dependence of BNQDs structural and optical properties on the growth process.

There are many things which can be added to this review article, however, considering it's length, it can be accepted in present form.

Response: We appreciate the positive comments. We are aware that there are numerous issues that can be considered, including applications. However, given the length of the article, we have decided to set aside some issues for future work.

Reviewer 3 Report

The work of Stagi et al is a review on the different techniques to synthesise BN-based materials, which is in the beginning complemented by reviewing different theoretical works devoted to the electronic, spectroscopic and optical properties.

The work is well written and understanding, and describes in a clear way the state of the art of the synthetic procedures for BN nanomaterials. Therefore, I don’t have any inconvenient to recommend the work to be published in the materials journal.

I have just one minor concern regarding the format of the work. The numbering of the sections is wrong. Every section starts with the number of “1” and each subsection with “1.1”. Authors have to fix that prior to publication.

Author Response

The work of Stagi et al is a review on the different techniques to synthesise BN-based materials, which is in the beginning complemented by reviewing different theoretical works devoted to the electronic, spectroscopic and optical properties.

The work is well written and understanding, and describes in a clear way the state of the art of the synthetic procedures for BN nanomaterials. Therefore, I don’t have any inconvenient to recommend the work to be published in the materials journal.

I have just one minor concern regarding the format of the work. The numbering of the sections is wrong. Every section starts with the number of “1” and each subsection with “1.1”. Authors have to fix that prior to publication.

Response: We thank the referee for his positive comments. We agree with the referee that there are some problems in the work format. Accordingly, we have fixed the numbering of the sections.